# Re-entrant DNA gels

Francesca Bomboi[1], Flavio Romano[2], Manuela Leo[1,3], Javier Fernandez-Castanon[1], Roberto Cerbino[4], Tommaso Bellini[4], Federico Bordi[1,5], Patrizia Filetici[6] & Francesco Sciortino[1,5]

DNA is acquiring a primary role in material development, self-assembling by design into complex supramolecular aggregates, the building block of a new-materials world. Using DNA nanoconstructs to translate sophisticated theoretical intuitions into experimental realizations by closely matching idealized models of colloidal particles is a much less explored avenue. Here we experimentally show that an appropriate selection of competing interactions enciphered in multiple DNA sequences results into the successful design of a one-pot DNA hydrogel that melts both on heating and on cooling. The relaxation time, measured by light scattering, slows down dramatically in a limited window of temperatures. The phase diagram displays a peculiar re-entrant shape, the hallmark of the competition between different bonding patterns. Our study shows that it is possible to rationally design biocompatible bulk materials with unconventional phase diagrams and tuneable properties by encoding into DNA sequences both the particle shape and the physics of the collective response.

[1] Dipartimento di Fisica, Sapienza Università di Roma, Piazzale A. Moro 2, 00185 Roma, Italy. [2] Dipartimento di Scienze Molecolari e Nanosistemi, Università Ca' Foscari di Venezia, Campus Scientifico, Edificio Alfa, via Torino 155, 30170 Venezia Mestre, Italy. [3] Dipartimento di Biologia e Biotecnologie C. Darwin, Sapienza Università di Roma, 00185 Roma, Italy. [4] Dipartimento di Biotecnologie Mediche e Medicina Traslazionale, Università degli Studi di Milano, via Fratelli Cervi 93, I-20090 Segrate (MI), Italy. [5] Istituto Sistemi Complessi-CNR, Sapienza Università di Roma, Piazzale A. Moro 2, 00185 Roma, Italy. [6] Istituto di Biologia e Patologia Molecolari-CNR, Sapienza Università di Roma, Piazzale A. Moro 2, 00185 Roma, Italy. Correspondence and requests for materials should be addressed to F.S. (email: francesco.sciortino@uniroma1.it).

The use of DNA oligonucleotides to build nano-sized particles[1,2], stemming from the Seeman's visionary intuition[3], is the result of a combination of factors that are uncommon in self-assembling molecular systems, such as the addressability of the binding sites, the steep temperature ($T$) dependence of the hybridization free energy and the high solubility in water. In addition, DNA hybridization is well characterized, both experimentally and theoretically[4], and DNA oligomers can nowadays be produced inexpensively in large quantities.

All these factors have contributed to the capacity of designing complex three-dimensional nanostructures, including nanostars (NSs) with tuneable number of arms[5–7], cubes[8], complex polyhedra[9,10] and tiles[11] to name a few. Constructs entirely made of DNA[12–14] have also been designed to act as actuators, switches[15] and logical gates. The majority of these applications have focused on the ability to design complex self-assembling shapes, whereas the collective behaviour of the resulting particles has been investigated to a much lesser extent[16,17]. Less explored is the use of DNA-made particles as model systems to experimentally verify theoretical predictions based on man-designed interaction potentials. Along this line, here we demonstrate the successful selection of short DNA sequences that spontaneously generate all-DNA particles with unconventional phase behaviour by encoding in the DNA sequences not only the required particle shape but also the desired and $T$-programmable collective properties of the resulting material.

As a proof-of-concept of the possibility to design DNA constructs mimicking peculiar model particles at the nanoscale, we focus on an interesting example of competition between different bonding patterns, recently addressed in a theoretical study of a mixture of tetravalent (A) and monovalent (B) patchy particles[18]. This model was originally designed to describe colloid agglomeration[19]. Such a system, theoretically and *in silico*[18], shows re-entrant gelation and a peculiar phase diagram in which the density of the coexisting dense and dilute phases approach each other on cooling[20].

Re-entrant condensation requires an additional mechanism that counteracts the standard driving force for phase separation. Examples can be found in (i) binary hydrogen-bonded fluids where translational entropy and orientational bonding entropy compete[21,22] producing close-loop coexistences; (ii) in micro emulsions, where the transition from cylinder to spherical micelle suppresses the standard phase separation[23,24]; (iii) in biological self-assembling systems such as G-actin, where polymerization depends non-monotonically on $T$[25,26]; or (iv) in dipolar hard spheres, where chain branching gives way to linear chains and rings on cooling[20,27]. Re-entrant behaviour has also been proposed as a technique to expand the crystallization sweet spot in DNA-coated colloidal systems[28] and applied in the experimental realization of an inter-particle effective potential, which changes from attractive to repulsive on cooling[29], and the control of the lattice constant of nanoparticle crystals[30].

We plan to recreate in the laboratory the patchy particles proposed in ref. 18 in which the additional mechanism inducing re-entrance originates from the competition between two possible bonding possibilities: AA bonds, for example, one of the four binding sites of particle A binding to one of the four sites of a distinct A particle, and AB bonds, for example, one of the four binding sites of particle A binding to the only site of a B particle. At high $T$, both AA and AB association is not present and the system behaves as a fluid of monomers. At very low $T$, the AB bonds become more favourable than the AA ones. As a result, the monovalent B particles associate with the tetravalent A monomers capping their arms and the system forms a fluid of diffusive $AB_4$ clusters (in which the four B particles saturate all bonding sites of particle A). In between, the AA bonds are predominant over the AB bonds and a spanning tetravalent network phase forms, that is, a highly viscous gel. Thus, theory suggests that under very specific conditions of the relative free-energy of the AA and AB interactions, the competition between these two bonding possibilities creates, in addition to a re-entrant phase-separation, a crossover from fluid to solid, to fluid again, providing a theoretical example of a material that can be hardened both on cooling and on heating. Such a system could be even designed to have a low viscosity (fluid) at ambient $T$ and an high one (gel) on heating at body $T$. Under these conditions, the material, when loaded with appropriate drugs or biological active molecules, is expected to improve drug delivery and crossing of biological barriers, extending drug-releasing times[31] and prolonging efficacy.

## Results

**System design**. Moving from *in silico* to the real world requires the ability to design the particles and their mutual interactions, and more importantly to produce them in bulk quantities, to be able to study their collective behaviour.

In addition, in the present case, the onset of the intermediate-$T$ gel phase requires designing AB bonds, which must be energetically stronger than the AA bonds (to be the relevant bonds at low $T$) but at the same time entropically more costly. This is necessary to force the AB bonds to become effectively active only at temperatures much smaller than the one that stabilizes the AA bonds. The implementation via physical interactions of this entropic stabilization is far from being trivial, but it can be made possible by exploiting the selectivity of DNA base pairs, as we will show below. The self-assembly strategy inherent in the build-up of DNA constructs has indeed a twofold advantage: not only it provides large amounts of identical particles but it also offers the opportunity to design and control both the strength of the interaction (controlled by the number of bases involved in the bonding) and the entropy cost of the hybridization process. Here we propose to use DNA tetravalent NS[6,7,13], previously investigated as building blocks for equilibrium gels[32], as the A particles. These double-stranded star structures with a flexible core spontaneously form at high $T$, when four properly designed single strands are mixed in solution (Fig. 1a,b). Each arm ends with a short self-complementary sequence of unpaired bases (six in ref. 7), which acts as a programmable sticky site, providing the gel connectivity (via the AA bonds). As shown previously[7], when only A particles are mixed in solution a physical gel forms (or melts) around the melting temperature of the sticky sequence. Please note that the self-complementary sequence has been designed to be short enough to avoid the formation of any stable secondary structure.

The most challenging part is the design of the competing sequences. Although it is clear that switching the AA bonds with AB bonds must lower the system free energy, it is necessary to invent a strategy such that the replacement takes place only at low $T$. To displace the AA bonds at low $T$ and thus melt the gel, exploiting the competing interaction paradigm, one needs to design appropriate B particles. A short sequence of DNA bases (non-self-complementary to avoid BB pairing) that is able to compete with the AA bond is the ideal candidate, provided that the AB bonds (i) form well below the $T$ at which the AA network is established and (ii) swiftly displace the AA bonds to avoid kinetic traps. To fulfill these requirements, we need to make the AA bond stronger and correspondingly increase the $T$ for the AA-gel formation. This is done by adding two extra bases to the self-complementary sequence originally chosen in ref. 7.

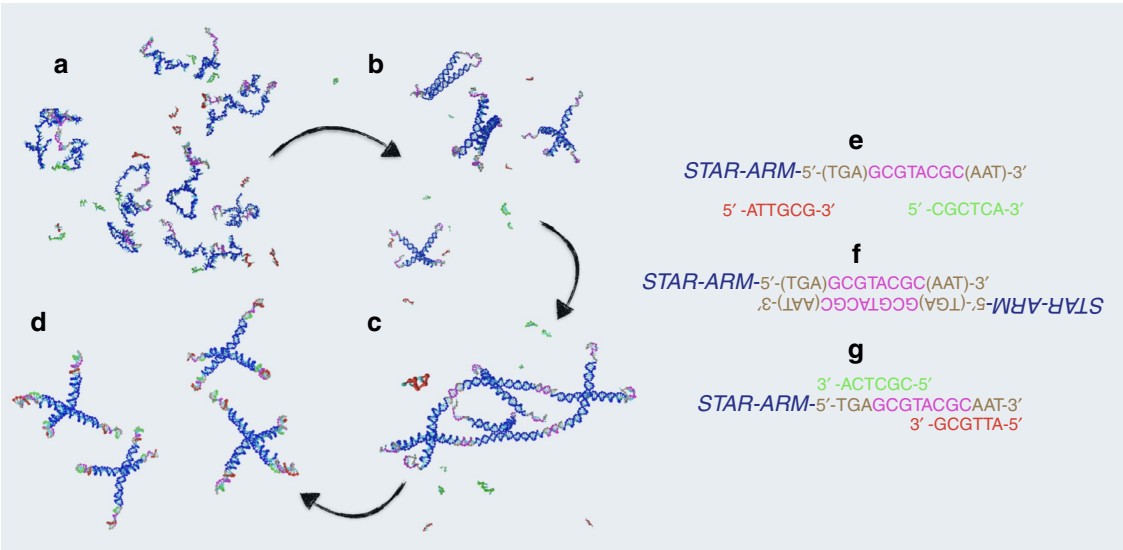

**Figure 1 | Sketch of the DNA sequences and of their aggregation state.** (**a**) Very high $T$ behaviour, where the four strands composing the A particle and the sequences defining the B particles are all not hybridized. (**b**) High $T$ behaviour, where the four strands have now hybridized to form the NSs, while the sequences defining the B particles have not yet hybridized. (**c**) Intermediate $T$ behaviour, where distinct NSs bind via the sticky sequences to form a gel, while the sequences defining the B particles have not yet hybridized. (**d**) Low $T$ behaviour, where the B particles have displaced the AA bonds originating free-floating $AB_4$ clusters. (**e**) Sequences composing the sticky ends. The self-complementary eight-base sequence that provides the AA bond (magenta) and the two unpaired three-base-long toehold sequences (brown) are noteworthy. (**f**) Sequence arrangement in the presence of an AA bond (8 base pairs in total). (**g**) Sequence arrangement in the presence of a full AB bond (12 base pairs in total).

To favour the bond swap process we also add at each end of the sticky sequence three further bases that will act as toehold[33–35] for the incoming displacing B sequence (Fig. 1e). Furthermore, we design two different short, single-stranded DNA sequences (the B particles), which are complementary to distinct ending parts of the newly designed NS sticky ends (Fig. 1f,g). The splitting of the B particles into two distinct sequences is crucial. It solves two of the major design challenges: (i) the requirement to avoid BB pairing. Indeed, being the A sticky sequence self-complementary to allow for AA bonding, a single B sequence competing for the same A sequence would also be self-complementary, becoming thus prone to self pairing. The use of two sequences competing for different regions of the AA bond solves the problem; (ii) the requirement to increase the entropy cost of forming an AB bond. As the entropy loss associated to the bonding of the two blocking oligomers is larger than the entropy loss associated to the bonding of one blocking oligomer of double length (owing to the additional freezing of the centre of mass degrees of freedom), this allows to lower the hybridization temperature of the AB bonds. Further details are provided in the Methods section and in Supplementary Discussion Sequence design.

The availability of accurate models of DNA hybridization thermodynamics[4] allows us to calculate, for the selected base sequences, the $T$-dependent probability of forming $AA$ ($p_{AA}$) and $AB$ ($p_{AB}$) bonds (Fig. 2). In the absence of B particles, $p_{AA}$ has the sigmoidal shape typical of the melting profile in two-state systems[36] (open/closed bonds). The addition of the B particles, competing for binding at the A sites, forces $p_{AA}(T)$ to go back to zero at low $T$, when the AB bonds have completely replaced the AA ones. It is important to stress that at intermediate $T$, when AB bonds are still rare, the value of $p_{AA}$ is well beyond the value for the percolation probability for tetrafunctional units ($p_{AA} = 1/3$ in mean field[37]). At this intermediate $T$, the system should thus be composed by a well-connected network of AA bonds (Fig. 1c) in the presence of floating B strands. At low $T$, $p_{AA} \rightarrow 0$, whereas simultaneously $p_{AB} \rightarrow 1$, completely melting the gel (Fig. 1d).

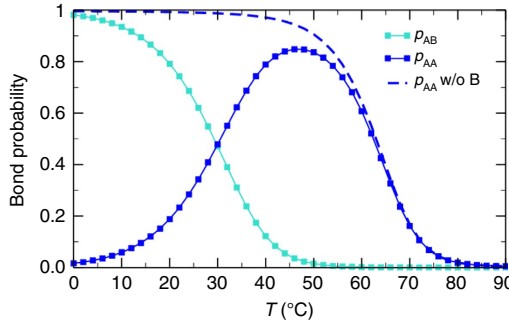

**Figure 2 | Temperature dependence of the probability of forming AA and AB bonds.** Evaluation of the gel AA ($p_{AA}$) and of the blocking AB ($p_{AB}$) bond probability as evaluated via the hybridisation free energies[4] implemented in the package NUPACK[48]. It is noteworthy that $p_{AA} + p_{AB} \neq 1$, as there is a small fraction of higher-order structures (for example, AAB). Four short B sequences of each of the two kinds are present for each NS, such that when all B particles are bonded, no AA bonds can exist. Each of the three strands (5'-TGA GCGTACGCAAT-3', 5'-ATTGCG-3', 5'-CGCTCA-3') has a concentration $8.5 \times 10^{-4}$ M and [NaCl] = 100 mM.

**Re-entrant phase behaviour.** The re-entrant behaviour and its associated structural change (from isolated stars (Fig. 1b) to a fully percolating network (Fig. 1c) and back to isolated stars (Fig. 1d)) have a signature in the system thermodynamics. The effective valence of the NSs, given by the number of arms available for gel formation, changes continuously from four at high $T$ to zero at low $T$ (when the sticky sequences are paired with the B strands), a realization of a $T$-dependent valence system[38]. Theoretical predictions originally proposed by Tlusty and Safran for dipolar hard colloids[20] and, more recently, for a certain class of patchy colloids[38,39] suggest that particles with a $T$-dependent valence are characterized by a re-entrant colloid-rich colloid-poor

phase separation, the analogue of the gas–liquid separation in atomic systems. We provide here evidence of this unconventional phenomenon—that is, the density of the colloid-rich phase coexisting with the colloid-poor phase progressively decreasing on cooling. By examining the samples in a wide range of NS concentrations and $T$, we map the conditions under which a meniscus, separating a denser sedimented phase from a dilute solution floating on top, is present. Figure 3a shows photographs of the samples with $c = 100\,\mu M$ at different $T$ with the indication of the meniscus level. Other concentrations are reported in Supplementary Discussion Phase diagram. The complete phase diagram is reported in Fig. 3b. These results show that at sufficiently low $c$, as the system is cooled from high $T$, it transforms from homogeneous to phase-separated and back to homogeneous. Above the NS concentration of $c^{max} \approx 120\,\mu M$, no sign of phase separation is detected and the system remains homogeneous for all $T$s. The remarkably low value of $c^{max}$ is a property of low-valence empty liquids[32,40], liquids formed by particles with a limited number of possible binding partners.

**Re-entrant dynamics.** In the homogeneous phase ($c \geq c^{max}$) we expect to observe the onset of the network and its low $T$ disruption. We quantify these crossovers via dynamic light scattering (DLS), a technique measuring the time dependence of the concentration fluctuations. Figure 4a,b display typical autocorrelation functions of the scattered fields. All correlations exhibit a two-step decay, a behaviour that reflects the two main processes governing the relaxation of concentration fluctuations: the rattling of the particles at fixed bonding pattern (the fast

decay) and the decorrelation of the network topology associated with the elementary bond breaking and reforming processes (the slow decay)[7]. As discussed in ref. 41, the slow decay time of the density fluctuation in tetravalent DNA gels, measured by DLS, correlates with the system bulk viscosity, suggesting a transition from a low-viscosity dilute solution to a high-viscosity gel around the melting temperature of the sticky-end sequence.

DLS results clearly indicate that in the homogeneous gel phase ($c > c^{max}$), the system remarkably slows down as $T$ is lowered

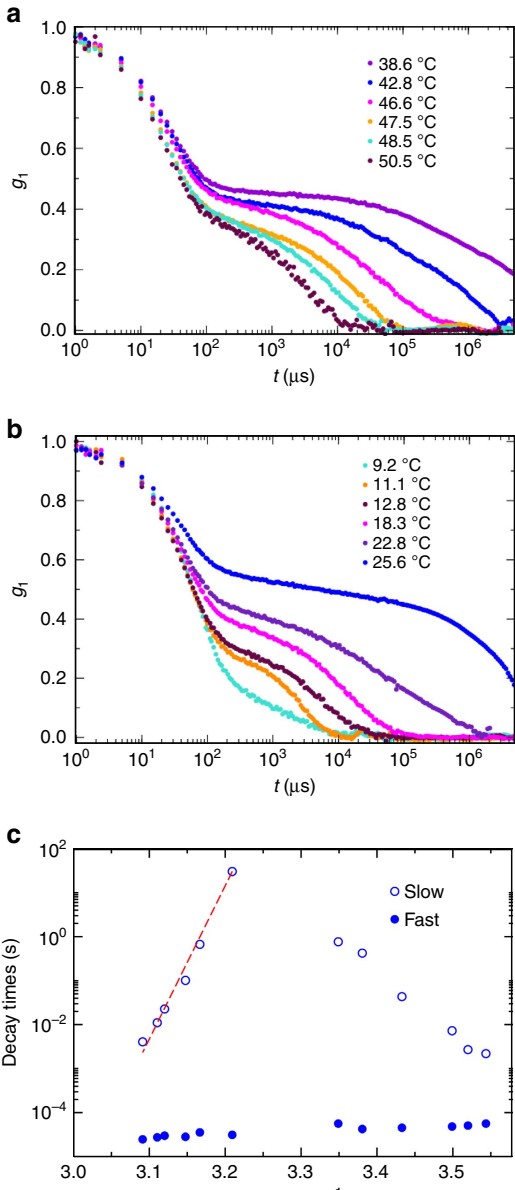

**Figure 4 | DLS correlation functions for different temperatures at $c = 213\,\mu M$.** (**a**) The region of gel formation where the decay time of the correlation functions progressively increases going beyond what can be experimentally measured ($\approx 10\,s$) when $T \approx 40\,^{\circ}C$. (**b**) The region of gel breaking, where the decay time of the correlation functions progressively decreases, indicating the melting of the network and the formation of $AB_4$ structures. (**c**) Temperature dependence of the average decay time as obtained by a stretched exponential fit of the slow decay as discussed in Supplementary Discussion Correlation function fit. The dashed red line is the best fitting curve of the slow decay characteristic time according to the Arrhenius law, with an activation energy of $160\,kcal\,mol^{-1}$.

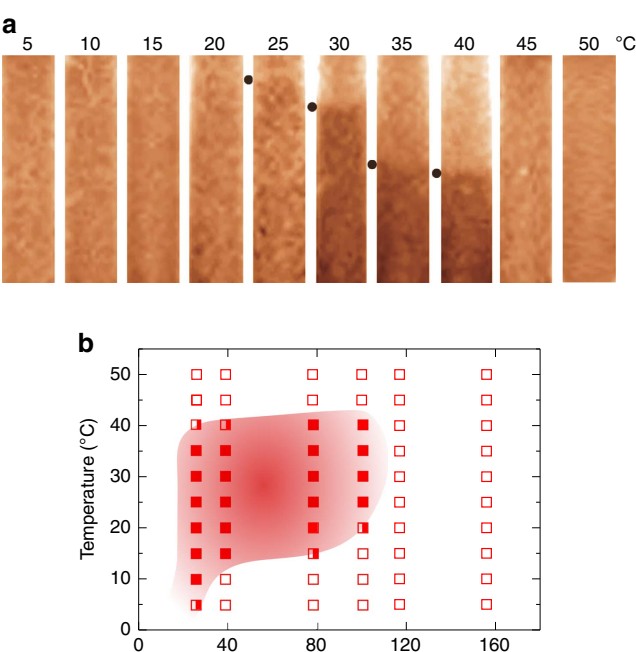

**Figure 3 | Phase diagram of the re-entrant DNA gel.** (**a**) Coloured photograph of the samples with $c = 100\,\mu M$ at different $T$ with the indication of the meniscus level (black points). The sample changes from homogeneous to phase separated to homogeneous again on cooling. Photographs of samples at different concentrations are reported in Supplementary Discussion Phase diagram. (**b**) Filled squares indicate phase separated state points, open squares indicate stable homogeneous solutions and semi-filled squares indicate borderline cases. The concentration is reported in $\mu M$ of NSs ($1\,mg\,ml^{-1} = 14.2\,\mu M$). The red shadow area qualitatively indicates the region of phase separation.

from 50 °C to 38 °C (Fig. 4a). In this high $T$ region, the gel forms and the AA bonds become longer and longer lived on cooling, consistent with the previously studied case of a one-component solution of A particles[7]. In addition, consistent with previous results[7,42], the slow decay time follows an Arrhenius slowing down (Fig. 4c) with a slope of $\approx 160$ kcal mol$^{-1}$, the expected value[42] for the chosen sticky sequence. At $T \approx 38$ °C, the dynamics becomes so slow that the correlation function extends beyond the experimentally accessible time scale (10 s).

The B particles start playing their role when the $T$ is lowered below 25 °C. The correlation functions become measurable again, but the slow decay time now decreases on lowering $T$ (Fig. 4b). This is due to the progressive substitution of the AA bonds with the AB bonds. Both at high and low $T$, far from percolation, the slow relaxation process can be properly represented by a stretched exponential function (see Supplementary Discussion Correlation function fit) and the resulting $T$-dependence of the slow decay time (Fig. 4c) shows a cusp-like non-monotonic behaviour, with a central $T$ window in which the relaxation time of the network is longer than the largest experimentally measurable value.

We conclude by noting that DNA gels are examples of thermo-reversible gels, in which binding arises from the hydrogen bonds between complementary base pairs, a clear realization of physical (as opposed to covalent) interactions. Supplementary Discussion Dynamic light scattering shows that the correlation functions shown in Fig. 4 are independent on the sample thermal history and remain identical over several thermo-reversible cycles.

## Discussion

In summary, our results show that the appropriate selection and use of DNA–DNA interactions makes it possible to generate bulk quantities of colloidal nanoparticles that can be used for designing and realizing self-assembled soft materials with unconventional properties. Specifically, tetravalent NSs exhibit in water a gel transition with a re-entrant phase behaviour[20], an evidence of competing thermodynamic mechanisms. On cooling, NSs aggregate, forming a persistent network gel structure[7,41,42]. The presence in solution of an appropriate amount of carefully designed monovalent DNA particles provides the additional mechanism opposing network formation. By design, at intermediate $T$ these monovalent particles are essentially inactive, but further lowering $T$ favours the bonding between monovalent particles and tetravalent ones, causing the progressive detachment of the latter from the network and the eventual disruption of the gel phase. The resulting material thus gels only in a limited $T$ range centred around human body values, making it possibly promising for pre-clinical and clinical use to facilitate medical treatments and therapies. In addition, we envision applications of such gels in vaccines[43] as antigen interacting adjuvant, in inflammation/arthritis[44] or bone tissue restoration[45] as support for cell migration and adhesion. Our results become particularly interesting for these applications in view of the fact that DNA interacts with RNA and/or proteins at specific oligonucleotides. It could also be suitable for (temperature controlled) gene therapy if adopted to interfere or reveal non-coding micro and long non-coding RNAs species[46,47], either in tissues or in blood stream, and use both as a diagnostic tool or as a selective microRNA trapping system. In more detail, the realization of a body-temperature DNA hydrogel, with a low viscosity (non-gel) at ambient temperature, has the potential to serve as a delivery tool to concentrate bioactive compounds in body tissues.

## Methods

**Sequence design.** The design of the DNA sequences is guided by oligonucleotide thermodynamics. Supplementary Discussion Sequence design provides a detailed explanation of the choices made regarding the length and base composition of the A and B sequences. Here we offer only a short description of the main motivation

and of the final choices. Star–star (AA) bonds need to be strong enough to be long-lived on experimental time scales at room $T$, but at the same time weak enough not to interfere with the assembly of the stars at high $T$. In addition, the sequence involved in the AA bond must be self-complementary and thus unfortunately prone to secondary structure formation. Hence, the sticky end sequence must be short enough to avoid the formation of hairpin structures. To satisfy all these requirements, we select a length of eight for the A sequence (Fig. 1e). The most difficult part is the design of the competing sequences. To force the replacement of the AA bonds with AB bonds only at low $T$, we select 12 base pairs for the AB bonds but we split these 12 base pairs into two different oligomers, each of them complementary to distinct parts of the NS sticky strand (Fig. 1f,g). This choice has a twofold reason: first, as the entropy loss associated to the bonding of the 2 blocking oligomers of length 6 is larger than the entropy loss associated to the bonding of 1 blocking oligomer of length 12, this allows to lower the hybridization temperature of the AB bonds; second, splitting the B particle into 2 strands allows us to select non-self-complementary B sequences, avoiding unwanted BB bonds. Practically, the base pair sequences were designed by scripting NUPACK[48] and looking for sequences that yield the following: (i) a high peak for $p_{AA}$ at intermediate $T$; (ii) a low value of $p_{AA}$ and a high value of $p_{AB}$ at low $T$; and (iii) at all $T$s, a minimal amount of unwanted higher-order structures and secondary structure within each strand. We automatically selected promising sequences out of tens of thousands of randomly generated ones and identified manually the optimal ones.

**Sample preparation.** The four DNA star-forming sequences were purchased from Integrated DNA Technologies with PAGE purification and used without any further treatment. PAGE guarantees a very high purity level, ensuring that the effective valence of the NSs is close to four. The DNA NS sample was obtained by mixing equimolar quantities of four 57-based oligomers. The single-strand concentrations were measured with a Thermo Scientific NanoDrop TM 1,000 Spectrophotometer.

5′-CTACTATGGCGGGGTGATAAAAACGGGAAGAGCATGCCCATCC A TGA GCGTACGC AAT-3′
5′-GGATGGGCATGCTCTTCCCGAACTCAACTGCCTGGTGATACG A TGA GCGTACGC AAT-3′
5′-CGTATCACCAGGCAGTTGAGAACATGCGAGGGTCCAATACCG A TGA GCGTACGC AAT-3′
5′-CGGTATTGGACCCTCGCATGAATTTATCACCCGCCATAGTAG A TGA GCGTACGC AAT-3′

These sequences were designed to hybridize into star-shaped four-armed structures[7] with a high yield, as confirmed by electrophoretic measurements on agarose gel[7]. The following two six-base-long single-stranded DNA oligomers (the B particles), complementary to different parts of the NS sticky end, were added in solution:

5′-ATTGCG-3′
5′-CGCTCA-3′

The six-base-long oligomers were purchased from Integrated DNA Technologies with standard desalting purification. In all samples, the ratio between the molar concentration of each competing sequence and NSs is fixed to four, such that when all B particles are bound, the system is composed of only $AB_4$ constructs. We dissolved DNA in NaCl solutions, fixing the total Na$^+$ concentration to 130 mM (assuming that each phosphate dissociates one cation). All DNA solutions were heated to 90 °C for 20 min (thermal annealing) and subsequently slowly cooled down with a rate of $\sim 10$ °C h$^{-1}$, to allow for optimal NS formation.

**Determination of the phase diagram.** To determine the phase diagram, glass micro-capillary pipettes (25 µl Microdispenser Drummond Scientific Company) were filled with $\sim 10$ µl of sample and flame sealed at both extremities. Six samples at increasing NSs concentrations $c$ (26, 39, 78, 100, 117 and 156 µM) were prepared. As the difference in $c$ between the two coexisting phases does not enable visual observation of the meniscus by the naked eyes, samples have been marked with 0.5 µl of ethidium bromide (EtBr) (respectively at EtBr concentrations of 0.03, 0.05, 0.10, 0.13, 0.15 and 0.20 mg ml$^{-1}$), inserted in the aforementioned home-made microcapillary cells and centrifuged overnight with a 5702 R Eppendorf Centrifuge (at 4,400 r.p.m.) at specific target $T$ to favour the faster establishment of a clear meniscus and to speed up the completion of the macroscopic phase separation. Centrifugation speeds up the kinetics, suggesting that the dilute-solution to gel re-entrant phase transition is an equilibrium transition. At these EtBr concentrations, there is $<1$ EtBr molecule per NS. After centrifugation, each capillary was promptly photographed with a Bio-Rad Chemidoc Mp System 1708280 to detect the presence and the position of a meniscus. In Supplementary Discussion Effects of Ethidium Bromide we report the melting curves for samples with and without EtBr, to confirm that EtBr increases the DNA melting temperature by a couple of degrees.

**Dynamic light scattering.** To perform DLS measurements, NMR cylindrical, borosilicate glass tubes (2.4 mm inner diameter) were filled with about 40 µl of sample, covered with about 30 µl of silicon oil, to avoid evaporation and condensation on the tube walls and flame sealed. DLS measurements were

performed on a custom light scattering setup designed to handle microlitre-sized samples and a 633 nm (Newport) He-Ne Laser source (17 mW) at a fixed scattering angle of 90°. A Brookhaven correlator provides the autocorrelation of the scattered light intensity that, via the Siegard relation, is transformed into the autocorrelation function of the scattered field $g_1(t)$. For each sample, we first thermalized the DNA solutions at 55 °C for 30 min and then slowly cooled it in steps of few degrees (always allowing an equilibration time of 30 min before measuring). For each temperature, we performed measurements lasting 30 min.

**Data availability.** The authors declare that all data supporting the findings of this study are available within the article and its Supplementary Information files.

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

## Acknowledgements

T.B., F. Bomboi, F. Bordi, R.C. and F.S. acknowledge support from MIUR through the PRIN action. J.F.C. and F.S. acknowledge support from ETN-COLLDENSE (H2020-MCSA-ITN-2014, Grant Number 642774). M.L., F. Bordi and F.S. acknowledge support from Sapienza. R.C. acknowledges funding from MIUR through the action Futuro in Ricerca, Project ANISOFT (RBFR125H0M). We thank F. Domenici for technical support with the UV measurements.

## Author contributions

F.R. designed the DNA sequences and the theoretical evaluations of the bond probabilities. F. Bomboi and M.L. prepared the samples. F. Bomboi and J.F.-C. performed DLS experiments. M.L. and F. Bomboi measured the phase diagram. All authors contributed to analyse the results and to write the article.

## Additional information

**Competing financial interests:** The authors declare no competing financial interests.

