## [Peer Review File · Nature Communications]

Reviewers' comments:

Reviewer #1 (Remarks to the Author):

This is an interesting paper that shows how DNA can be used as a basis for producing varied types of self assembled structures with novel properties such as re-entrant sol-gel transitions. The paper can be suitable for the interdisciplinary audience of Nature Communications after the authors address the following:

1. More motivation is needed for the audience of Nature Communications. Why is the re-entrant sol-gel transition of wide interest? Is there any practical implementation?
2. In a similar vein, the authors should explain a bit more of how DNA can give varied self-assembled structures with additional examples and how they compare to the colloidal examples that the authors quote near the bottom of page 2.
3. I suggest that instead (or in addition) to the use of the term liquid-gas transition that the authors use the term sol-gel transition since this is in effect what they claim.
4. Is there any more direct experimental evidence such as electron microscopy of the connected gel phase? What about rheology?
5. The physical origin of the re-entrant behavior as far as I understand it in general terms is that at high temperature, chain formation is prevented by the fact that the AA association probability falls to very small values. At low temperature, the "monomers" of the gel phase associate with the "B" particles that cap them and prevent chain and network formation. In between, the network phase is stable. In other words, at high temperatures, the chains comprising the network are not the stable structure, while at low temperatures, a competing phase transition (AB association) prevents gel formation. A good analogy for this that should also be referenced is network formation in micellar systems that form worm like microemulsions/micelles and networks in a limited temperature (or composition) range. In those systems, the networks have indeed been imaged. The competition in that system between structures relates to the cylinder (that is the basic network "monomer") to sphere transition (that is analogous to the AB, capped assembly that prevents network formation) at low temperatures or cylinder to lamellar transition at high temperature. This is discussed phenomenologically in a PRL on re-entrant behavior by Menes et al. in 1995 and is even closer to the DNA system than the magnetic analogy. The PRL deals with the dependence of the re-entrant behavior on composition but the shape transitions themselves are governed by temperature as pointed out in the paper by Zilman et al. in Langmuir 2004. Fig. 2 of that paper, shows the temperature dependence of the two phase region and the low temperature spherical phase that pre-empts the network formation. At high temperatures, lamellar or bicontinuous phases can occur, again pre-empting the network phase (since the cylinders are again unstable), similar to the plot of PAA in Fig. 2 of the present manuscript which shows that chain formation (analogous to cylinders in the surfactant systems) is only favored in a limited temperature range. This Langmuir paper also discusses the kinetics of various transitions which may also be relevant to the DNA system.
6. The authors should give the physical explanation of the non-monotonic behavior of PAA as a function of temperature in Fig. 2. It is preferable if this can be done in a "scaling" manner without detailed atomistic/molecular calculations. One normally expects such interactions to be monotonic in temperature, perhaps due to Boltzmann factor arguments.

Is the non-monotonic behavior on the atomistic/molecular scale related to hydrogen bonding which shifts its nature as a function of temperature or to other such effects. Note that on the atomistic/molecular level re-entrant phase behavior occurs for hydrogen bonded systems: see papers by Walker and Vause and Goldstein and Walker.

Reviewer #2 (Remarks to the Author):

In this manuscript, the authors described their efforts on the realization of Safran-like re-entrant phase behavior experimentally with DNA modular self-assembly. The power of DNA self-assembly was demonstrated to tune the uncommon phase diagram, with structural melting at both high and low temperatures. This work is too specialized. Regarding the novelty, DNA-based hydrogel has been reported in many papers. Although the phase behavior seems interesting, controlling phase transition of hydrogel using DNA strands displacement was reported before. Centrifugation was required to realize phase separation and no mechanical properties of formed hydrogel were demonstrated. I cannot recommend publication of this manuscript.

Reviewer #3 comments

Hydrogels, in particular, the newly invented DNA-based hydrogels, are fascinating materials from both theoretical and practical point of views; studies on phase behavior of hydrogel systems are important for achieving more fundamental understandings as well as for ultimate applications. In this paper, the authors attempted to rationally design the relevant DNA sequences in order to, experimentally (the key component of this paper), control the phase behavior and thus achieve a one-pot DNA hydrogel that melted both on heating and on cooling. This unusual behavior might provide, claimed by the authors, significant potential for biomedical applications.

Major issues:

1. While this study was intriguing and interesting, in particular with the clever design of DNA sequences, the experimental work itself was a bit thin in characterizing the formed DNA hydrogels; the study can be further enhanced with more experiments. After all, this paper self-highlighted the experimental realization as the key. Just as the authors adopted the experimental design, thermodynamically closed system would be suitable for study of phase transition behavior as well, by using, say DSC, to obtain the enthalpy changes and the heat of phase transition, etc. The additional experiments would be important to better understand the transition energy changes in thermodynamic process, and thus provide more experimental support for thermodynamic analysis of competing interaction based on claimed base bonding. In addition, as concluded by the authors, the gel itself would be used in biomedical applications such as drug delivery. Thus it would be important to know how encapsulated drugs would affect phase behavior (using a few drugs as model).

2. It is also important for the authors to clearly articulate the differences and distinguish the presented work from the previously published studies ("Phase behavior and critical activated dynamics of limited-valence dna nanostars" (PNAS 2013, 110, 15633)). In this manuscript, the authors controlled the phase behavior through adjusting the bonding pattern with the aid of competitive sequences and eventually achieved a non-monotonic phase transition process. The idea was clever, however, it appeared that the work was an extension and expansion of previous work (this notion underscores the aforementioned needs to conduct more experimental characterizations).

3. It would be interesting to know whether or not the temperature-dependent re-entrant phase behavior is reversible. If not, then how to recover the DNA hydrogel? The reversibility might be critical for the applications of such materials.

4. A major problem of this manuscript is the inflated, non-supported conclusion based on pure speculation. ("We conclude noting that biocompatible gels are promising for several medical treatments, in vaccines [25] as novel antigen interacting adjuvants, in inflammation/arthritis [26] or bone tissue restoration [27] as supports for cell migration and adhesion."). There is no evidence whatsoever in the presented work to suggest that such DNA hydrogels can even be used in biomedical applications or better than many other non-DNA hydrogels.

Technical issues:

1. In Figure 3, for the low concentration region (20 μm or lower) and for the high concentration region (120 μm or higher), will the DNA nano-stars form a hydrogel? Why was there no phase separation? How to explain this phenomenon from the view of thermodynamics?

2. Also, in the references section, are the URL links necessary for refs.15 and 22?

(please note that citations are reported at the end. The reference numbers of this letter differ from the ones of the manuscript.)

Reviewer # 1's

We report in italic the Reviewer comments and in red the new text

This is an interesting paper that shows how DNA can be used as a basis for producing varied types of self assembled structures with novel properties such as re-entrant sol-gel transitions. The paper can be suitable for the interdisciplinary audience of Nature Communications after the authors address the following:

We thank the Reviewer for stating that our work is interesting and, in principle, suited for Nature Communications. In the following we explain how we have addresses all points one by one.

- *1. More motivation is needed for the audience of Nature Communications. Why is the re-entrant sol-gel transition of wide interest? Is there any practical implementation?*

The previous manuscript had been directly transferred from Nature Physics and it was thus suffering from space limits. The revised version has been now formatted following the Nature Communications editorial policies, allowing for a significantly higher number of words. We have thus capitalised on this opportunity to expand the motivation part of the manuscript, following the Reviewer suggestion.

We have added in the introduction the following sentence: **Such a system could be even designed to be easily injectable (fluid) at ambient T and gelling on heating at body T . In this conditions, the material, when loaded with appropriate drugs or biological active molecules, is expected to improve drug-delivery and crossing of biological barriers, extending time of drug release [6] and prolonging efficacy.**

and in the conclusions: **The resulting material thus gels only in a limited T range centered around human body values, making it very promising for pre-clinical and clinical use to facilitate medical treatments and therapies. Also, we envision application of such gel in vaccines [1] as novel antigen interacting adjuvant, in inflammation/arthritis [2] or bone tissue restoration [3] as support for cell migration and adhesion. Our results become particularly interesting for these applications in view of the fact that DNA interacts with RNA and/or proteins at specific oligonucleotides. It could also be suitable for (temperature-controlled) gene therapy if adopted to interfere or reveal non-coding micro and long-non-coding RNAs species [4, 5] either in tissues or in blood stream and used both as a diagnostic tool or as a selective miRNAs trapping system. In more detail, the realisation of a body temperature DNA-hydrogel, injectable at ambient temperature, and interacting with bioactive**

molecules/drugs/antigens has the potential to serve as a novel delivery tool to concentrate bioactive compounds in body tissues.

- 2. *In a similar vein, the authors should explain a bit more of how DNA can give varied self-assembled structures with additional examples and how they compare to the colloidal examples that the authors quote near the bottom of page 2.*

We thank the Reviewer for suggesting us to discuss the variety of DNA self-assembled structures and how DNA particles can be produced to resemble colloidal systems at the nanoscale. We have now added the following sentence: **All these factors have contributed to the capacity of designing complex 3D nanostructures, including nanostars (NS) with tunable number of arms [7, 8, 9], cubes [10], complex polyhedra [11, 12], tiles [13] to name a few. Constructs entirely made of DNA [14, 15, 16] have also been designed to act as actuators as well as logical gates. The majority of these applications have focused on the ability to design and self-assemble complex shapes but only to a lesser extent to the collective behaviour of the resulting particles [17, 18]. An even lesser explored avenue is the use of DNA-made particles as model systems to experimentally verify theoretical predictions based on man-designed interaction potentials. Along this line, here we demonstrate the successful selection of short DNA sequences that spontaneously generate all-DNA particles with unconventional phase behaviour, by encoding in the DNA sequences not only the required particle shape, but also the desired and T -programmable material collective properties.**

- 3. *I suggest that instead (or in addition) to the use of the term liquid-gas transition that the authors use the term sol-gel transition since this is in effect what they claim.*

The present system shows simultaneously phase-separation and sol-gel transition. The "liquid" phase (at intermediate T) is characterized by the presence of a network of bonds and hence can also be referred to as a gel. We have followed the Reviewer suggestion and paid particular attention to the use of the words "liquid" and "gel".

- 4. *Is there any more direct experimental evidence such as electron microscopy of the connected gel phase? What about rheology?*

The viscosity of tetravalent nanostar gels, formed with *only* A particles, has been previously evaluated (see ESI of Biffi et al Soft Matter, 11, 3132, 2015) by measuring the brownian diffusion time of particles of size $0.34 \mu\text{m}$ dissolved in the DNA solution, via the Stokes-Einstein relation. We reproduce here the published result.

In that study, we have shown that the viscosity perfectly correlates with the slow relaxation time measured in the DNA solution. Hence, the reported slow-time decays (Fig. 4) provides an accurate although indirect measurement of the temperature dependence of the viscosity. We have now added a sentence in the manuscript to call attention on this previously established connection. **As discussed in Ref. [19],**

FIG. S3: Temperature dependence of the viscosity η as extracted from the diffusion coefficient of microparticles (filled red dots) within a sample of concentration c_1 . The green line indicates the viscosity of water η_w . The dotted line has the same slope (i.e. the same ΔH) as the one in Fig. 3-b of the main text.

the slow decay time of the density fluctuation in tetravalent DNA gels, measured by DLS, strongly correlates with the system bulk viscosity.

Some of us are planning to perform laser tweezer micro-rheology in the near future in collaboration with Dr. Erika Eiser in Cambridge. We hope to be able to present results for the frequency dependence of the complex shear modulus in a future publication.

- 5. *The physical origin of the re-entrant behavior as far as I understand it in general terms is that at high temperature, chain formation is prevented by the fact that the AA association probability falls to very small values. At low temperature, the "monomers" of the gel phase associate with the "B" particles that cap them and prevent chain and network formation. In between, the network phase is stable. In other words, at high temperatures, the chains comprising the network are not the stable structure, while at low temperatures, a competing phase transition (AB association) prevents gel formation.*

A good analogy for this that should also be referenced is network formation in micellar systems that form worm like micro-emulsions/micelles and networks in a limited temperature (or composition) range. In those systems, the networks have indeed been imaged. The competition in that system between structures relates to the cylinder (that is the basic network "monomer") to sphere transition (that is analogous to the

AB, capped assembly that prevents network formation) at low temperatures or cylinder to lamellar transition at high temperature. This is discussed phenomenologically in a PRL on re-entrant behavior by Menes et al. in 1995 and is even closer to the DNA system than the magnetic analogy. The PRL deals with the dependence of the re-entrant behavior on composition but the shape transitions themselves are governed by temperature as pointed out in the paper by Zilman et al. in Langmuir 2004. Fig. 2 of that paper, shows the temperature dependence of the two phase region and the low temperature spherical phase that pre-empts the network formation. At high temperatures, lamellar or bicontinuous phases can occur, again pre-empting the network phase (since the cylinders are again unstable), similar to the plot of PAA in Fig. 2 of the present manuscript which shows that chain formation (analogous to cylinders in the surfactant systems) is only favoured in a limited temperature range. This Langmuir paper also discusses the kinetics of various transitions which may also be relevant to the DNA system.

The physical origin of the re-entrant behaviour in general terms is indeed the one that the Reviewer condenses in the first lines of this point. It is a competing transition, between structurally different configurations. The cylinder-rod competing transition in micro emulsions, the linear chains-branched clusters in Safran's model of dipolar hard-spheres as well as the re-entrant phase behavior which occurs for hydrogen bonded systems discussed in point 6 are examples of competing phase transition. In the revised version of the manuscript we discuss these systems (in addition to the ones which were already cited) and the analogies with the present one. The opportunity to expand the text offers us the possibility to comment these further examples.

Following the advices, we have added the sentence:

Re-entrant condensation requires an additional mechanism that counteracts the standard driving force for phase separation. Examples can be found in binary hydrogen-bonded fluids where translational entropy and orientational bonding entropy compete [20, 21] producing close-loop coexistences, in micro emulsions, where the transition from cylinder to spheric micelle suppresses the standard phase-separation [22, 23], in biological self-assembling systems like G-actin, where polymerisation depends non monotonically on T [24, 25] or in dipolar hard spheres where chain branching gives way to linear chains and rings on cooling [26, 27].

We have also better detailed the physical origin of the competition in the specific case, by adding:

We plan to recreate in the laboratory the patchy particles model proposed in Ref. [28] in which the additional mechanism inducing re-entrance originates from the competition between two possible bonding possibilities: AA bonds, e.g. one of the four binding sites of particle A binding to one of the four sites of a distinct A particle, and AB bonds, e.g. one of the four binding sites of particle A binding to the only

site of a B particle. At high T both AA and AB association is not present and the system behaves as a fluid of monomers. At very low T , the AB bonds become more probable than the AA ones. As a result, the monovalent B particles associate with the tetravalent A monomers capping them and the system forms a fluid of diffusive AB_4 clusters (in which the four B particles saturate all bonding sites of particle A). In between, the AA bonds are predominant over the AB bonds and a spanning tetravalent network phase forms, i.e. a highly viscous gel. Thus theory suggests that, under very specific conditions of the relative free-energy of the AA and AB interactions, the competition between these two bonding possibilities creates, in addition to a re-entrant phase-separation, a cross-over from fluid to solid to fluid again, providing a theoretical example of a material that can be hardened both on cooling and on heating.

- 6. *The authors should give the physical explanation of the non-monotonic behavior of PAA as a function of temperature in Fig. 2. It is preferable if this can be done in a "scaling" manner without detailed atomistic/molecular calculations. One normally expects such interactions to be monotonic in temperature, perhaps due to Boltzmann factor arguments. Is the non-monotonic behavior on the atomistic/molecular scale related to hydrogen bonding which shifts its nature as a function of temperature or to other such effects. Note that on the atomistic/molecular level re-entrant phase behavior occurs for hydrogen bonded systems: see papers by Walker and Vause and Goldstein and Walker.*

We have rewritten the discussion of the data presented in Fig. 2 to focus more on the physical explanation of the non-monotonic behavior of p_{AA} as a function of temperature. As correctly pointed out by the Reviewer, one normally expects such interactions to be monotonic in temperature. The non-monotonic behavior is indeed the hallmark of the competition between two different bonding opportunities: the AA bond and the AB bond. We have also modified Fig.2 to include the p_{AA} prediction in the case in which the B particles are not present (green line in Fig.2). In this case, p_{AA} retains the expected monotonic temperature dependence.

The availability of accurate models of DNA hybridisation thermodynamics [29] allows us to calculate, for the selected base sequences, the T -dependent probability of forming AA (p_{AA}) and AB (p_{AB}) bonds (Fig. 2). In the absence of B particles, p_{AA} has the sigmoidal shape typical of the melting profile in two-state systems [30] (open/close bonds). The addition of the B particles, competing for binding at the A sites, forces $p_{AA}(T)$ to go back to zero at low T , when the AB bonds have completely replaced the AA ones.

We have also added a sentence that better explains the constraints that need to be satisfied in the design of the DNA sequences

In addition, in the present case, the onset of the intermediate- T gel phase requires

designing AB bonds which must be energetically stronger than the AA bonds (to be the relevant bonds at low T) but at the same time entropically more costly. This is necessary to force the AB bonds to become effectively active only at temperatures much smaller than the one that stabilizes the AA bonds. The implementation via physical interactions of this entropic stabilization is far from being trivial, but it can be made possible by exploiting DNA base pairing selectivity, as we will show below.

and expanded the section describing the challenging design of the B particles:

The most challenging part is the design of the competing sequences. While it is clear that switching the AA bonds with AB bonds must lower the system free energy, it is necessary to invent a strategy such that the replacement takes place *only* at low T . To displace the AA bonds at low T and thus melt the gel, exploiting the competing interaction paradigm, one needs to design appropriate B particles. A short sequence of DNA bases (non-self-complementary to avoid BB pairing) that is able to compete with the AA bond is the ideal candidate, provided that the AB bonds (i) form well below the T at which the AA network is established and (ii) swiftly displace the AA bonds to avoid kinetic traps. To fulfil these requests, we need to make the AA bond stronger and correspondingly increase the T for the AA-gel formation. This is done by adding two extra bases to the self-complementary sequence originally chosen in Ref. [9]. To favour the bond swap process we also add at each end of the sticky sequence three further bases which will act as toehold [31] for the incoming displacing B sequence (Fig. 1(e)). Furthermore, we design two different short single-stranded DNA sequences (the B particles) which are complementary to distinct ending parts of the newly designed NS sticky ends (Fig. 1(f-g)). The splitting of the B particles in two distinct sequences is crucial. It solves two of the major design problems: (i) the request to avoid BB pairing. Indeed, being the A sticky sequence "palindromic" to allow for AA bonding, a single B sequence, competing for the same A sequence would also be palindromic, becoming thus prone to self pairing. The use of two sequences competing for different regions of the AA bond solves the problem; (ii) the request to increase the entropy cost of forming an AB bond. Since the entropy loss associated to the bonding of the two blocking oligomers is larger than the entropy loss associated to the bonding of one blocking oligomer of double length (due to the additional freezing of the center of mass degrees of freedom), this allows lower the hybridization temperature of the AB bonds.

Reviewer #2

In this manuscript, the authors described their efforts on the realization of Safran-like re-entrant phase behavior experimentally with DNA modular self-assembly. The power of DNA self-assembly was demonstrated to tune the uncommon phase diagram, with structural melting at both high and low temperatures. This work is too specialized. Regarding the novelty, DNA-based hydrogel has been reported in many papers. Although the phase behavior seems interesting, controlling phase transition of hydrogel using DNA strands displacement

was reported before. Centrifugation was required to realize phase separation and no mechanical properties of formed hydrogel were demonstrated. I cannot recommend publication of this manuscript.

Perhaps we have not been able to clearly convey the novelty of our work, e.g. the demonstration that it is possible to design DNA particles that interact as assumed in sophisticated particle models, to provide experimental verification of theoretical studies by encoding in the DNA sequences the physics of the collective behaviour we like to generate.

In the revised version we now clearly state: **An even lesser explored avenue is the use of DNA made particles as model systems to experimentally verify theoretical predictions based on man-designed interaction potentials. Along this line, here we demonstrate the successful selection of short DNA sequences that spontaneously generate all-DNA particles with unconventional phase behaviour, by encoding in the DNA sequences not only the required particle shape, but also the desired and T -programmable material collective properties.**

We agree 100 per cent that DNA-based hydrogels have been reported in many previous publications. But the formation of a DNA hydrogel is *not* the focus of our work. The focus is the demonstration that, exploiting the power of DNA-self assembly, it is possible to create a DNA system that gels on heating. This has never been reported before. Concerning the statement "controlling phase transition of hydrogel using DNA strands displacement was reported before", we are not aware of any study in which phase transition of hydrogels are controlled with DNA strands displacement. A precise reference to support the Reviewer's statements would have been most helpful. We are familiar with the work (our previous reference [13]) in which strand displacement of DNA coated colloids was exploited to control the interaction potential between colloids, but this is very different from our *melting-on-heating* hydrogel system. Finally, we have clarified that centrifugation is only requested to favour the faster establishment of a clear meniscus by writing **centrifuged overnight with a 5702 R Eppendorf Centrifuge (at 4400 rpm) at specific target T ... to favour the faster establishment of a clear meniscus and to speed-up the completion of the macroscopic phase separation.** The simple fact that all the samples outside the phase separation remain homogeneous even after centrifugation is the best evidence that phase separation is not induced by centrifugation.

Reviewer #3

Hydrogels, in particular, the newly invented DNA-based hydrogels, are fascinating materials from both theoretical and practical point of views; studies on phase behavior of hydrogel systems are important for achieving more fundamental understandings as well as for ultimate applications. In this paper, the authors attempted to rationally design the relevant DNA sequences in order to, experimentally (the key component of this paper), control the phase behavior and thus achieve a one-pot DNA hydrogel that melted both on heating and on cooling. This unusual behavior might provide, claimed by the authors, significant potential for biomedical applications.

Major issues:

- 1. *While this study was intriguing and interesting, in particular with the clever design of DNA sequences, the experimental work itself was a bit thin in characterizing the formed DNA hydrogels; the study can be further enhanced with more experiments. After all, this paper self-highlighted the experimental realization as the key. Just as the authors adopted the experimental design, thermodynamically closed system would be suitable for study of phase transition behavior as well, by using, say DSC, to obtain the enthalpy changes and the heat of phase transition, etc. The additional experiments would be important to better understand the transition energy changes in thermodynamic process, and thus provide more experimental support for thermodynamic analysis of competing interaction based on claimed base bonding. In addition, as concluded by the authors, the gel itself would be used in biomedical applications such as drug delivery. Thus it would be important to know how encapsulated drugs would affect phase behavior (using a few drugs as model).*

We thank the Reviewer for finding our study intriguing and interesting. We can only agree with the Reviewer that further experiments and technical applications are highly welcome and we are sure our work will generate a significant interest in the scientific community. Still, we can not provide these experimental results at the present time. We stress that it is not the goal of our study to prove the biomedical applications. Our goal is to prove that the refined design of the DNA sequences allows us to generate a hydrogel that melts both on heating and on cooling. This is what we proved. We claim, in the conclusions, that the possibility to create a biocompatible gel, fluid at ambient temperature and solid at body temperature has certainly the potentiality to be of relevance for biomedical applications and we listed few fields where such applications could be relevant. But this is clearly marked as "envisioning", not as proving. And, we stress once more, this is not the goal of the manuscript.

Concerning DSC measurements, this is possibly not the best experimental technique to detect a reentrant behaviour in DNA-made particles. Indeed the trick we develop is based on base switching, from the AA to the AB bonds. The number of DNA base pairs progressively increases on cooling and hence the enthalpy will not show any

re-entrance. We have calculated, based on the accurate Santalucia [29] expression, the theoretically expected T -dependence of the enthalpy. The results are reported in the following figure. While the AA component of the enthalpy (blue line) does show a re-entrance, the experimentally accessible total enthalpy (black line) is monotonically decreasing due to the progressive formation of AB bonds (red line) that compensate for the displaced AA bonds. We agree that a DSC experiment could provide an interesting additional characterisation of the system we have introduced, marking the melting temperatures of the AA and AB bonds, but it would not be crucial for proving that the proposed design of DNA particles does generate a re-entrant gel.

- 2. *It is also important for the authors to clearly articulate the differences and distinguish the presented work from the previously published studies ("Phase behavior and critical activated dynamics of limited-valence dna nanostars" (PNAS 2013, 110, 15633)). In this manuscript, the authors controlled the phase behavior through adjusting the bonding pattern with the aid of competitive sequences and eventually achieved a non-monotonic phase transition process. The idea was clever, however, it appeared that the work was an extension and expansion of previous work (this notion underscores the aforementioned needs to conduct more experimental characterizations).*

In the revised version we stress further the difference between the two studies. In the old PNAS work we investigated the valence dependence. With valence we indicate the number of sticky arms of the nano-star (NS). We showed that on decreasing the valence, the "gas-liquid" phase diagram shifts progressively to smaller concentration, opening up larger concentration regions in which the system forms an equilibrium gel.

As it was written in the manuscript "Here we propose to use DNA tetravalent NS [8, 15, 9], previously investigated as building blocks for equilibrium gels [32], as the A particles". So, yes, we capitalise on the expertise previously developed to build the four-functional particles. But we had to modify the ending sequence of each arm to encode for the re-entrant mechanism. The revised version expands and articulates

the differences between the presented work and the PNAS.

To fulfil these requests, we modify the previously used binding sequence of the AA particles. Specifically, to increase the T for the AA-gel formation we make the AA bond stronger. This is done by adding two extra bases to the self-complementary sequence previously chosen in Ref. [9]. To favour the bond swap process we also add at each end of the sticky sequence three further bases which will act as toehold [31] for the incoming displacing B sequence (Fig. 1-(e)). The presence of the toehold sequence allows for the formation of intermediate states in which the B particles can bind to the arm of the NS before starting the bond swapping process. In this way the activation barrier for opening the AA bond and substituting it with an AB bond is strongly reduced, minimising the possibility of kinetic traps.

- 3. *It would be interesting to know whether or not the temperature-dependent re-entrant phase behavior is reversible. If not, then how to recover the DNA hydrogel? The reversibility might be critical for the applications of such materials.*

We thank the Reviewer for giving us the possibility to clarify this issue. The material is fully reversible and reproducible. We have repeated the experiments both on cooling (starting from high T where the system is fluid) and on heating (starting from low T , where the system is again fluid) always recovering the same results. The reversibility of DNA gels formed by NS (only AA interactions) had been already tested (see Fig. 3a in Ref. [19]). The following figure shows the comparison between the correlation function measured on cooling and on heating for the re-entrant gel, both above and below the intermediate T gel region. After cooling and before heating the system has been kept to 5°C for several hours. As shown in the figure, the results are completely independent on the thermal history.

We have now added the following sentence in the manuscript and an additional paragraph in the supplementary information material, reporting the history-independent

results shown above.

We conclude by noting that DNA gels are examples of thermo-reversible gels, in which binding arises from the hydrogen bonds between complementary base-pairs, a clear realisation of physical (as opposed to covalent) interactions. The S.I. shows that the correlation functions shown in Fig. 4 are indeed independent on the sample thermal history.

- 4. *A major problem of this manuscript is the inflated, non-supported conclusion based on pure speculation. ("We conclude noting that biocompatible gels are promising for several medical treatments, in vaccines [25] as novel antigen interacting adjuvants, in inflammation/arthritis [26] or bone tissue restoration [27] as supports for cell migration and adhesion."). There is no evidence whatsoever in the presented work to suggest that such DNA hydrogels can even be used in biomedical applications or better than many other non-DNA hydrogels.*

As we had alluded before, we wanted only to call attention of our biologist, material scientist, chemist colleagues that this material has potential application in biomedicine and related fields. Possibly the word "We conclude noting .. are promising ..." gave the wrong impression. We have now replaced it with "... possibly promising ...", hoping that this clarifies our goal. We also note that Reviewer #1 asked us to motivate more possible applications. Only for this reason we do not eliminate the final sentence completely. But if the Editor will suggest us to do it, we will certainly eliminate these additional motivations, which do not reflect the main focus of the manuscript.

Technical issues:

- 1. *In Figure 3, for the low concentration region ($20 \mu\text{m}$ or lower) and for the high concentration region ($120 \mu\text{m}$ or higher), will the DNA nano-stars form a hydrogel? Why was there no phase separation? How to explain this phenomenon from the view of thermodynamics?*

The question of the Reviewer clearly shows us that we gave for granted the knowledge of the thermodynamic phase behaviour in systems with temperature dependent valence (Ref. [26, 33]). The possibility to expand the length of the text offers us now the chance to discuss in more detail the meaning of the presence of a phase separation in a limited region of the phase diagram.

In the "Reentrant phase behavior" section we have added the following sentences:

Phase separation in T -dependent valence systems is characterized by a limited region of instability in the $T - c$ plane, that progressively shrinks toward vanishing concentrations when the valence approaches two [26, 33]. According to theoretical predictions, in the coexisting high concentration phase, the system is able to form a stable network structure whose restructuring time is controlled by the lifetime of the

inter particle bonds (the AA bonds in our case). The coexisting low concentration phase is instead formed by a fluid of finite-size clusters. The absence of phase separation above the coexisting concentration (e.g. the formation of equilibrium gels) can be traced back to the geometrical possibility to form, without generating regions of strong density fluctuations, a fully bonded structure [34].

and later on

Experimental sensitivity limits ourselves to $c > 26 \mu\text{M}$, preventing us from clearly detecting the low-concentration boundary of the coexistence curve, where the system is expected to be composed by isolated NS. The important result is that above the NS concentration of $c^{max} \approx 120 \mu\text{M}$, no sign of phase separation is detected and the system remains homogeneous for all T s. The remarkably low value of c^{max} is a property of low-valence "empty liquids" [32, 35], liquids formed by particles with a limited number of possible binding partners. Thus, for $c > c^{max}$ we expect to observe a region in which the system percolates such that, if the lifetime of the AA bonds is sufficiently large, a persistent gel is formed.

- 2. Also, in the references section, are the URL links necessary for refs.15 and 22?

We have eliminated the URL in the old refs.15 and 22, following the Reviewer advice.

References

- [1] Singh, A. & Peppas, N. A. Hydrogels and scaffolds for immunomodulation. *Adv. Mater.* **26**, 6530–6541 (2014).
- [2] Goindi, S., Narula, M. & Kalra, A. Microemulsion-based topical hydrogels of tenoxicam for treatment of arthritis. *AAPS PharmSciTech* 1–10 (2015).
- [3] Huebsch, N. *et al.* Matrix elasticity of void-forming hydrogels controls transplanted-stem-cell-mediated bone formation. *Nat. Mater.* **14**, 1269–1277 (2015).
- [4] Shu, Y. *et al.* Stable rna nanoparticles as potential new generation drugs for cancer therapy. *Adv. Drug Deliver. Rev.* **66**, 74–89 (2014).
- [5] Ferracin, M. & Negrini, M. Micromarkers 2.0: an update on the role of micrnas in cancer diagnosis and prognosis. *Expert Rev. Mol. Diagn.* **15**, 1369–1381 (2015).
- [6] Almeida, J. C. *et al.* A biocompatible hybrid material with simultaneous calcium and strontium release capability for bone tissue repair. *Mater. Sci. Eng.: C* **62**, 429–438 (2016).
- [7] Seeman, N. C. Nucleic acid junctions and lattices. *J. Theor. Biol.* **99**, 237–247 (1982).
- [8] Li, Y. *et al.* Controlled assembly of dendrimer-like dna. *Nat. Mater.* **3**, 38–42 (2004).

- [9] Biffi, S. *et al.* Phase behavior and critical activated dynamics of limited-valence dna nanostars. *Proc. Natl. Acad. Sci.* **110**, 15633–15637 (2013).
- [10] Chen, J. & Seeman, N. C. Synthesis from dna of a molecule with the connectivity of a cube. *Nature* **350**, 631–633 (1991).
- [11] Goodman, R. P. *et al.* Rapid chiral assembly of rigid dna building blocks for molecular nanofabrication. *Science* **310**, 1661–1665 (2005).
- [12] He, Y. *et al.* Hierarchical self-assembly of dna into symmetric supramolecular polyhedra. *Nature* **452**, 198–201 (2008).
- [13] Wei, B., Dai, M. & Yin, P. Complex shapes self-assembled from single-stranded dna tiles. *Nature* **485**, 623–626 (2012).
- [14] Pinheiro, A. V., Han, D., Shih, W. M. & Yan, H. Challenges and opportunities for structural dna nanotechnology. *Nat. Nanotech.* **6**, 763–772 (2011).
- [15] Roh, Y. H., Ruiz, R. C., Peng, S., Lee, J. B. & Luo, D. Engineering dna-based functional materials. *Chem. Soc. Rev.* **40**, 5730–5744 (2011).
- [16] Qian, L. & Winfree, E. Scaling up digital circuit computation with dna strand displacement cascades. *Science* **332**, 1196–1201 (2011).
- [17] Um, S. H. *et al.* Enzyme-catalysed assembly of dna hydrogel. *Nat. Mater.* **5**, 797–801 (2006).
- [18] Zheng, J. *et al.* From molecular to macroscopic via the rational design of a self-assembled 3d dna crystal. *Nature* **461**, 74–77 (2009).
- [19] Biffi, S. *et al.* Equilibrium gels of low-valence dna nanostars: a colloidal model for strong glass formers. *Soft matter* **11**, 3132–3138 (2015).
- [20] Vause, C. A. & Walker, J. S. Effects of orientational degrees of freedom in closed-loop solubility phase diagrams. *Phys. Lett. A* **90**, 419–424 (1982).
- [21] Walker, J. S. & Vause, C. A. Lattice theory of binary fluid mixtures: Phase diagrams with upper and lower critical solution points from a renormalization-group calculation. *J. Chem. Phys.* **79**, 2660–2676 (1983).
- [22] Menes, R., Safran, S. & Strey, R. Reentrant phase separation in microemulsions. *Phys. Rev. Lett.* **74**, 3399 (1995).
- [23] Zilman, A., Safran, S., Sottmann, T. & Strey, R. Temperature dependence of the thermodynamics and kinetics of micellar solutions. *Langmuir* **20**, 2199–2207 (2004).

- [24] Matthews, J. N. *et al.* The polymerization of actin: extent of polymerization under pressure, volume change of polymerization, and relaxation after temperature jumps. *J. Chem. Phys.* **123**, 074904 (2005).
- [25] Dudowicz, J., Douglas, J. F. & Freed, K. F. Equilibrium polymerization models of re-entrant self-assembly. *J. Chem. Phys.* **130**, 164905 (2009).
- [26] Flusty, T. & Safran, S. A. Defect-induced phase separation in dipolar fluids. *Science* **290**, 1328–1331 (2000).
- [27] Rovigatti, L., Tavares, J. M. & Sciortino, F. Self-assembly in chains, rings, and branches: a single component system with two critical points. *Phys. Rev. Lett.* **111**, 168302 (2013).
- [28] Roldán-Vargas, S., Smallenburg, F., Kob, W. & Sciortino, F. Gelling by heating. *Sci. Rep.* **3** (2013).
- [29] SantaLucia, J. A unified view of polymer, dumbbell, and oligonucleotide dna nearest-neighbor thermodynamics. *Proc. Natl. Acad. Sci.* **95**, 1460–1465 (1998).
- [30] Hill, T. L. *An introduction to statistical thermodynamics* (Courier Corporation, 2012).
- [31] Zhang, D. Y. & Winfree, E. Control of dna strand displacement kinetics using toehold exchange. *J. Am. Chem. Soc.* **131**, 17303–17314 (2009).
- [32] Bianchi, E., Largo, J., Tartaglia, P., Zaccarelli, E. & Sciortino, F. Phase diagram of patchy colloids: Towards empty liquids. *Phys. Rev. Lett.* **97**, 168301 (2006).
- [33] Russo, J., Tavares, J. M., Teixeira, P. I. C., Telo da Gama, M. M. & Sciortino, F. Reentrant phase diagram of network fluids. *Phys. Rev. Lett.* **106**, 085703 (2011).
- [34] Zaccarelli, E. Colloidal gels: equilibrium and non-equilibrium routes. *J. Phys.-Condens. Mat.* **19**, 323101 (2007).
- [35] Ruzicka, B. *et al.* Observation of empty liquids and equilibrium gels in a colloidal clay. *Nat. Mater.* **10**, 56–60 (2011).

REVIEWERS' COMMENTS:

Reviewer #1 (Remarks to the Author):

This revised version seriously and comprehensively addresses all of my concerns and I recommend publication in Nature Communications. There are a few very minor points that the authors should still consider. The Editors can check these and there is no need for further review on my part:

1. In the conclusions, where they say injectable, they should say "low viscosity (non-gel) fluid" and use the same term when they write "injectable" in the revised introduction.
2. To this end, when they use the term "liquid-gas" they should clarify this as "liquid (gel) - gas (dilute solution)". Everything here is in solution and word "gas" could lead to confusion.
3. In the added sentence "As discussed in Ref. [19].." the authors could be more explicit and instead "strongly correlates with" write suggests a transition from a low-viscosity dilute solution to a high viscosity gel at a temperature of Also, is Fig. S3 reproduced in the SI of the current paper? If not, they should refer to the specific fig. in Ref. 19.
4. Where the authors use the word "request" in the responses to both Ref. 1 and Ref. 3, I suggest that they instead use the word "require".
5. In the response to Referee 2, the authors may want to add that the fact that centrifugation only speeds up the kinetics suggests that the dilute solution to gel re-entrant phase transition is one of equilibrium.

Reviewer #2 (Remarks to the Author):

The authors discovered unconventional phase behaviors from well-studied DNA hydrogelation, which is interesting. Prior to the acceptance by Nature communications, there are several concerns to be addressed.

1. How many cycles do the thermo-reversible gel-sol transitions take place?
2. There are no mechanical properties indicated, as requested by the reviewer 2. Mechanical strength is a very important property for hydrogel. Also, since the hydrogel presents a novel phase behavior, the author should provide microscopic images (e.g.) of the cross-section of the hydrogel.
3. The triggered switching of DNA-based hybrid hydrogel via strand displacement has been well reported. The references should be cited.
 - a. Journal of Materials Research. 2005, 20, 1
 - b. Macromolecules. 2005, 38, 1535.
 - c. J. Am. Chem. Soc., 2015, 137, 15723-15731

It was also reported the primer produced by the strand displacement reactions was involved in the formation of the hydrogel (Nature Nanotechnology, 2012, 7, 816-820).

Reviewer #3 (Remarks to the Author):

The revised ms is much better written now. While I still believe that more experiments are needed to make a stronger case on the practical impact of this ms (after all, the ms is for Nature Communications as oppose to a more specialized journal), I am relatively satisfied with the current version in which comparing with the previous version, the arguments and the novelty are more solidly presented. I really hope that their all-DNA gels would behave as designed and as predicted when combined with drugs (including other DNA/RNA molecules). Only time will tell.

Reviewer # 1's

We thank Reviewer # 1 for recommending publication in Nature Communications. Following his/her advices we have

- 1. substituted "injectable" with "low viscosity (non-gel) fluid" both in the introduction and in the conclusions.
- 2. eliminated the words "gas" and "liquid" everywhere except in the sentence "the analogue of gas-liquid separation in atomic systems".
- 3. substituted "strongly correlates" with "correlates with the system bulk viscosity, suggesting a transition from a low-viscosity dilute solution to a high viscosity gel around the melting temperature of the sticky-end sequence."
- 4. substituted the word "request" with the word "require" everywhere in the text.
- 5. added "Centrifugation speeds up the kinetics, suggesting that the dilute solution to gel re-entrant phase transition is an equilibrium transition."

Reviewer #2

We thank Reviewer #2 for her/his additional suggestions

- 1. We have clarified that the thermo-reversible gel-sol transitions can be repeated for months with the same efficiency. More precisely, when discussing the reversibility of the transition, we have added "identical over several thermoreversible cycles"
- 2. We agree with the Reviewer that mechanical properties are important for hydrogels and we plan to perform laser tweezer micro-rheology in the near future. We hope to be able to present results for the frequency dependence of the complex shear modulus in a future publication.
- 3. We thank the Reviewer for bringing to our attention interesting references. We have added them in the revised version.

Reviewer #3

We thank Reviewer #3 for appreciating our efforts in revising the manuscript and we are glad to read that she/he is relatively satisfied with the current version, in which the arguments and the novelty are more solidly presented. We can reassure she/him that we will continue working on this material to provide evidence that it behaves as designed and as predicted when combined with drugs (including other DNA/RNA molecules). We agree that "only time will tell".